# Correlation between Ferromagnetic Layer Easy Axis and the Tilt Angle of Self Assembled Chiral Molecules

**DOI:** 10.3390/molecules25246036

**Published:** 2020-12-20

**Authors:** Nir Sukenik, Francesco Tassinari, Shira Yochelis, Oded Millo, Lech Tomasz Baczewski, Yossi Paltiel

**Affiliations:** 1Applied Physics Department and the Center for Nanoscience and Nanotechnology, The Hebrew University of Jerusalem, Jerusalem 91904, Israel; nir.sukenik@mail.huji.ac.il (N.S.); shira.yochelis@mail.huji.ac.il (S.Y.); 2Department of Chemical and Biological Physics, Weizmann Institute of Science, Rehovot 76100, Israel; francesco.tassinari@gmail.com; 3Racah Institute of Physics and the Center for Nanoscience and Nanotechnology, The Hebrew University of Jerusalem, Jerusalem 91904, Israel; milode@mail.huji.ac.il; 4Magnetic Heterostructures Laboratory, Institute of Physics, Polish Academy of Sciences, Al. Lotnikow 32/46, 02-668 Warszawa, Poland; bacze@ifpan.edu.pl

**Keywords:** chiral induced spin selectivity effect, self-assembled monolayer, enantioselective adsorption, magnetic anisotropy, magnetic thin films

## Abstract

The spin–spin interactions between chiral molecules and ferromagnetic metals were found to be strongly affected by the chiral induced spin selectivity effect. Previous works unraveled two complementary phenomena: magnetization reorientation of ferromagnetic thin film upon adsorption of chiral molecules and different interaction rate of opposite enantiomers with a magnetic substrate. These phenomena were all observed when the easy axis of the ferromagnet was out of plane. In this work, the effects of the ferromagnetic easy axis direction, on both the chiral molecular monolayer tilt angle and the magnetization reorientation of the magnetic substrate, are studied using magnetic force microscopy. We have also studied the effect of an applied external magnetic field during the adsorption process. Our results show a clear correlation between the ferromagnetic layer easy axis direction and the tilt angle of the bonded molecules. This tilt angle was found to be larger for an in plane easy axis as compared to an out of plane easy axis. Adsorption under external magnetic field shows that magnetization reorientation occurs also after the adsorption event. These findings show that the interaction between chiral molecules and ferromagnetic layers stabilizes the magnetic reorientation, even after the adsorption, and strongly depends on the anisotropy of the magnetic substrate. This unique behavior is important for developing enantiomer separation techniques using magnetic substrates.

## 1. Introduction

It is well established by now that the probability of electrons passing through chiral molecules depends on the momentum of the charge and its spin. This effect has been named as the Chiral Induced Spin Selectivity (CISS) effect [1,2,3]. A relatively new aspect of this effect was recently discovered, and it is related with the interaction between chiral molecules and ferromagnetic substrate with perpendicular anisotropy. These interactions create two complementary phenomena. The first is magnetization reorientation of the ferromagnetic (FM) film where the molecules are adsorbed, even without any external magnetic or electric field [4]. The second, is a substantial differentiation of adsorption rate between the two enantiomers of the same molecule - one enantiomer adsorbs preferentially when the magnetic dipole is pointing up, whereas the other adsorbs faster for the opposite alignment of the magnetization - thus enabling chiral separation [5]. Both phenomena occur when the FM has an out of plane (OoP) easy axis for magnetization, thus the easy axis has a prominent component aligned with the bonding angle of the molecular layer (roughly 30–60° from the normal to the surface [6]). It was also discovered that when a chiral molecular monolayer is exposed to an external magnetic field the tilt angle of the monolayer can be altered [7]. In this work, we present the influence of the FM easy axis direction and magnetic coercivity, on both the magnetization reorientation of the FM, due to the chiral molecules adsorption, and on the tilt angle of the self-assembled monolayer (SAM). For this purpose, SAMs were selectively adsorbed on an epitaxial Au/Co/Au nanostructure with a wedge shape of the ferromagnetic Co layer giving a thickness gradient. Such ferromagnetic Co layers have an OoP easy axis for the thinner films and an in plane (IP) easy axis for the thicker films. Magnetic atomic force microscopy (MFM) measurements of the thin film areas with adsorbed molecules were performed, each providing a correlated magnetization map (in the out of plane direction) and a respective topographic image of the adsorbed areas that is directly connected to the tilt angle of the monolayer (as defined in Figure 1). The measurements were performed either with or without external magnetic field during the adsorption process. A clear correlation is found between the easy axis of the FM film and the tilt angle of the monolayer for the same adsorption conditions. The molecules tend to tilt by a larger angle, in respect to a plane normal (laying down), when the FM easy axis is IP for both with and without an external field (see illustration in Figure 1).

Furthermore, we demonstrate that when the FM layer easy axis is OoP, the magnetization reorientation, due to the adsorbed chiral monolayer, occurs even if there is an applied magnetic field during the adsorption process, leading to the conclusion that the magnetization reorientation occurs also after the initial bonding event. These results deepen our understanding of the interaction between FMs and chiral molecules, and may contribute to the implementation of this interaction in spintronic and magnetic devices.

## 2. Results

Figure 2 shows the MFM results measured on a selectively adsorbed SAM of L- alpha helix polyalanine (L-AHPA) chiral molecules (as described in the Materials and Methods section) deposited on a gradient Co substrate without an external magnetic field. It is clear from the topography (Figure 2a,c) that there are well-defined adsorption areas (squares of 1 × 1 micron) of the SAM. As expected, as it was measured previously [4], there is a perpendicular magnetization component in the OoP easy axis part of the Co wedge corresponding directly to the areas with adsorbed SAM (Figure 2d). On the other hand, the IP easy axis area shows only magnetic domains without order related to the molecular adsorption (Figure 2b). The MFM tip is magnetized perpendicular to the surface and is unable to detect in plane magnetization component (except for domain edges). This does not mean that there is no local magnetization.

Next, we repeated the above experiment with applied constant magnetic field during the adsorption. Figure 3 presents the results when the adsorption was done under an applied external magnetic field of 3000 Gauss, oriented parallel to the sample surface (in-plane). This magnetic field is much larger than the coercive field of the FM layer (roughly a few hundreds of Gauss, as can be seen in Appendix A). An external field of 3000 Gauss, applied in-plane, is strong enough to force the magnetization vector into the sample plane throughout the sample. Again, from the topography images (Figure 3a,c) it is clear that there are areas with adsorbed SAM. The in-plane easy axis area shows no magnetic signal detected by MFM (Figure 3b) even though there is certainly a magnetization in the IP direction (due to the strong applied external magnetic field). Surprisingly, the OoP easy axis area does show perpendicular magnetization component (see Figure 3d) in the adsorbed areas even though an IP magnetic field of 3000 Gauss was applied during the entire adsorption process, meaning the magnetization ordering occurred after the molecule’s adsorption event.

When adsorbing a SAM on a film with OoP easy axis, under an OoP magnetic field (also ~3000 Gauss) (Figure 4), it is clear that there is a magnetic signal when the north pole of the magnet is facing the sample (Figure 4b), and no magnetic signal when the south pole is facing the sample (Figure 4d). This does not mean that there is no perpendicular magnetization component in the south pole case, but rather that in the entire area of the ferromagnetic layer measured, the magnetization is aligned in the same direction in which case, there is no magnetic contrast that can be detected with MFM. Since the L-AHPA molecules used in the experiment reorient the magnetization of the FM layer in a particular direction (as was shown in Figure 2c and also in ref. [4]), then if this direction is the same as the orientation of magnetization in the FM layer forced by the external magnetic field, then there will be no change in the magnetization orientation due to the chiral molecules adsorption, hence no contrast in magnetization will emerge (see Figure 4d). If the FM magnetization direction induced by the molecular adsorption is opposite to the orientation of magnetization in the FM layer forced by the external magnetic field, then there will be a difference in the magnetization orientation between areas with adsorbed molecules and without it, and thus a contrast can be visualized in MFM measurements (see Figure 4c).

To reveal the correlation between the easy axis and the bond tilt angle of the SAM, a numerical analysis of the topographic measurements, described in the Materials and Methods part, was done. Table 1 summarizes the values of different tilt angles for the sample areas with IP and OoP easy axis with and without an applied external magnetic field. Two findings are noticeable in these results. The first is that for both when adsorption was done with and without an external magnetic field the tilt angle of the SAM on the OoP easy axis films is smaller than that on the film with IP easy axis. The second is that for the wedge area with IP easy axis there is a difference in tilt angles affected by the external IP magnetic field. The tilt angle of the SAM is slightly bigger for adsorption with magnetic field assistance than without the field. On the other hand, in the OoP easy axis area there is an overlap between the tilt angles for the cases when external field is applied IP (i.e., perpendicular to the easy axis) or not. Similar trends for the difference between IP and OoP easy axis are apprehended using a polarization modulation–infrared reflection–absorption mode in FTIR (see Appendix A) as was done elsewhere [8].

## 3. Discussion

The results presented above reflect a clear correlation between the easy axis direction of the FM layer and the tilt angle of the adsorbed molecules. This correlation links the selective adsorption of molecules with certain chirality on a perpendicularly magnetized substrate [5], and the magnetization reorientation caused by the SAM adsorption process [4]. The charge redistribution throughout the molecule during the adsorption process is accompanied by spin polarization parallel to the chiral axis of the molecule. The polarized spins at a molecule’s edge site then interact differently with a given magnetization orientation of the FM layer, creating a preference to spins polarization parallel to the magnetization direction. Thus, for an IP easy axis case the molecules tend to tilt more in order to create the parallel spin interaction, while if the easy axis is OoP, the molecules tilt is reduced. When an external magnetic field is applied in the IP easy axis direction the tilt angle increases, compared to the zero-field adsorption case. However, if the external magnetic field is perpendicular to the FM layer easy axis the tilt angle is unchanged; this may imply that the change in tilt angle continues after the adsorption event, when there is a magnetic relaxation of the magnetization towards the easy axis. Since tilt angle alignment occurs simultaneously with the ferromagnetic layer magnetization reorientation a competition between the two effects occurs, resulting in a slightly different tilt angles for different initial conditions, both FM easy axes direction and applied external magnetic field.

It is important to notice one more surprising result — the fact that magnetization, detected by MFM, appeared in the OoP easy axis wedge area, even when the adsorption was realized under an external IP magnetic field of 3000 Gauss. Also, when an OoP magnetic field is applied during the adsorption process on an OoP easy axis Co wedge area, there is no magnetic signal when the south magnetic field is applied, in contrast to the case when the north magnetic field is applied. Both of these results can be explained if we assume that the magnetization reorientation in FM layer is not realized only during the chemical bonding process, but rather it is a gradual and prolonged process. As described before [9], charge polarization, caused due to a change in the electric field or electrochemical potential, is accompanied by spin polarization along the chiral molecule axis. This spin ’dipole’ in the molecule splits the energies of electrons (or holes) penetrating the molecule, depending on their spin state (spin parallel or anti-parallel to the spin ’dipole’ of the molecule). This energy splitting leads to a spin blockade effect, which results in an induced magnetization reversal in the FM layer. Thus, even though the FM layer was magnetized in the direction of the applied external magnetic field during the adsorption process, the molecules induce an energetic preference to a certain spin orientation even after adsorption. Other results that support this effect and suggest that the magnetization ordering occurs on a long timescale were measured using nitrogen vacancies (NV) centers in diamond as a sensitive magnetometer and will be published soon elsewhere.

## 4. Materials and Methods 

The magnetic substrate used here is an epitaxial nanostructure grown by the Molecular Beam Epitaxy (MBE) method at the Institute of Physics of the Polish Academy of Sciences in Warsaw. The sample configuration is:

Al_2_O_3_(0001)/Pt 5 nm/Au 20 nm/Co wedge/Au 5nm, where the Co thickness changes linearly along the substrate edge as a wedge in the range from 1.14 nm to 2.85 nm, thus changing the magnetization direction from an OoP easy axis to an IP easy axis for a critical thickness of 2.1 nm (as shown by polar magneto optic Kerr effect measurement shown in Appendix A). The gold layer on top acts both as a capping layer to prevent oxidation of the Co layer and a surface for covalent bonding of the thiol group of the L-AHPA, thus enabling the SAM process.

Markers were patterned on the substrate using standard photolithography, then selective adsorption areas were patterned in Polymethyl Methacrylate (PMMA) electron resist using E-beam lithography. The patterned adsorption areas were squares of size 1 × 1 μm^2^ spaced apart by 1 μm between each square (as can be seen in Figure 2a,c, Figure 3a,c and Figure 4a,c).

The molecular layer was prepared using a SAM method. An organic thiolated α helix Poly-L-alanine: [H]-CAAAAKAAAAKAAAAKAAAAKAAAAKAAAAKAAAAK-[OH], where C, A, and K are Cysteine, Alanine, and Lysine amino acids, respectively, (purchased from Sigma-Aldrich) was adsorbed onto the substrate using several steps. First, the devices were left in absolute ethanol for 20 min before they were immersed into a 1mM ethanol solution of the organic molecule for 4 h. This procedure allows the SAM to form a homogeneous, closely packed single layer of chiral molecules. The excess of the organic molecules are removed from the surface by washing the sample with ethanol [10]. The PMMA is then removed with acetone and rinsed again with ethanol. All the adsorption process was done under nitrogen environment.

Topography and magnetic measurements were done using the MFM option of a Ntegra–NT–MDT Modular SPM apparatus with a magnetic tip (μmasch HQ: NSC36/Co-Cr/Al BS tip with a length of 90 μm and a force constant of 2 N/m) magnetized perpendicular to the surface of the sample. The MFM measurements were performed applying the double-path technique where in the second (MFM data collection) path the tip was at a distance of 25 nm from the surface (as in [4]).

Height of the monolayer was analyzed numerically by taking multiple cross sections and creating a histogram of heights then fitting the histogram to 2 Gaussians and taking the height of the monolayer as the distance between the two Gaussian centers.

Tilt angles were then calculated by simple trigonometry.

## 5. Conclusions

Our results mainly show two effects. The first one is the correlation between the easy axis of a FM layer and the bonding angle of adsorbed chiral α helix L-Polyalanine molecules. The second one is the fact that magnetization ordering in the FM layer, after chiral molecules are adsorbed on its surface, occurs on a long timescale, and persists well after the molecules are bonded to the surface.

Both of these effects constitute a binding stone between two known phenomena caused by the CISS effect; magnetization reorientation, induced by chiral molecules adsorption, and enantioselective adsorption, induced by FM layer magnetization orientation. Hence they both serve as important tools for a better and profound understanding of the CISS effect. This understanding is necessary for the implementation and application of the CISS effect for both magnetic and spintronic devices and for chiral separation uses and biological homochirality recognition.

## Figures and Tables

**Figure 1 molecules-25-06036-f001:**
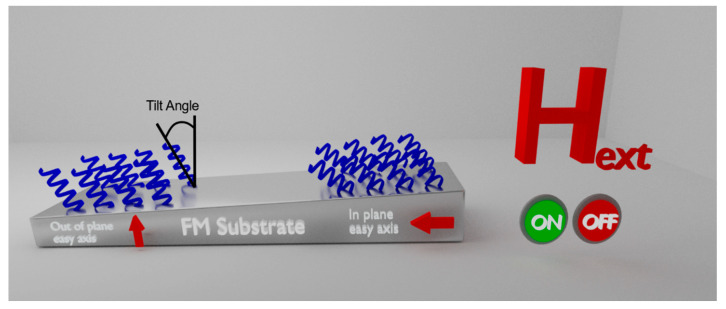
Schematic illustration of the experimental setup. A gradient thickness of ferromagnetic Co substrate with a transition between in plane to out of plane easy axis on which alpha helix L-polyalanine chiral molecules are selectively adsorbed. The tilt angle (defined in the figure) and magnetization are measured using MFM. The experiment is repeated when an external magnetic field (3000 Gauss) is applied both in plane and out of plane during the adsorption process.

**Figure 2 molecules-25-06036-f002:**
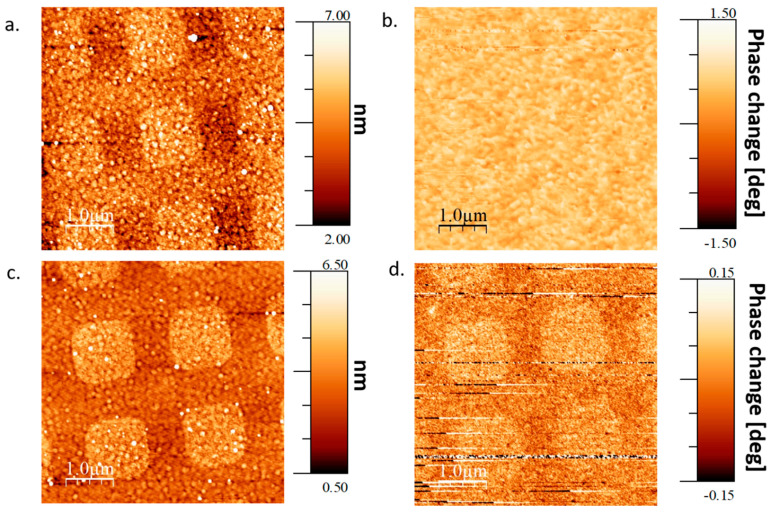
Topography and magnetization maps for L-AHPA SAM adsorption without applied external magnetic field. (**a**). Topography of the SAM in IP easy axis wedge area. (**b**). Magnetization phase of the SAM in IP easy axis Co wedge area. (**c**). Topography of the SAM in OoP easy axis wedge area. (**d**). Magnetization phase of the SAM in OoP easy axis wedge area. It is clear that there is a perpendicular magnetization component in the OoP easy axis wedge areas while it is absent in the IP region.

**Figure 3 molecules-25-06036-f003:**
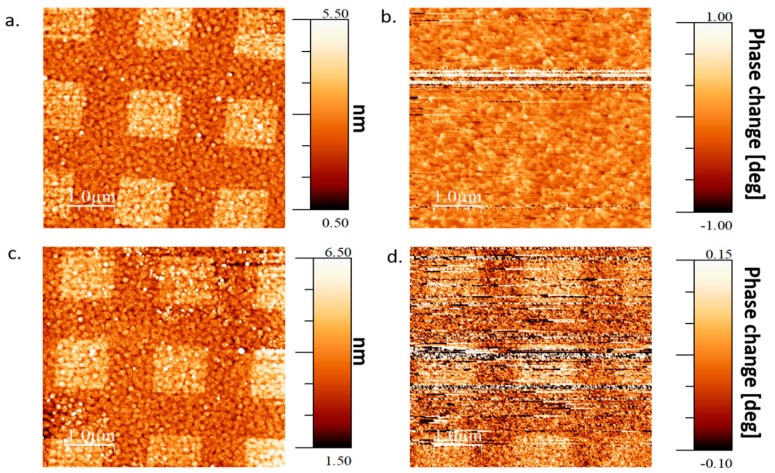
Topography and magnetization with applied IP external magnetic field. (**a**). Topography of a L-AHPA SAM in an IP easy axis Co wedge area. (**b**). Magnetization of same area as shown in (**a**). (**c**). Topography of a SAM in an OoP easy axis wedge area. (**d**). Magnetization of the SAM in the same OoP easy axis wedge area as in (**c**).

**Figure 4 molecules-25-06036-f004:**
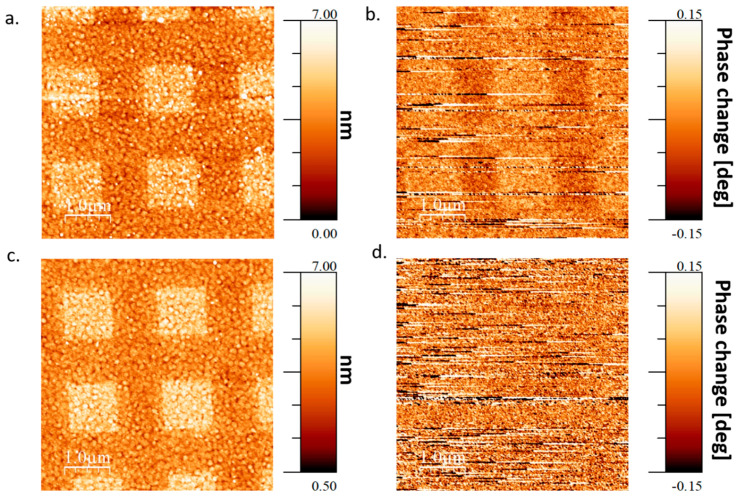
Topographic and magnetic image of OoP easy axis film with an applied OoP external magnetic field during the adsorption process. (**a**). Topography of a SAM for adsorption when the north pole of the magnet is facing the sample. (**b**). Magnetic image measured in parallel with the topographic image presented in (**a**). (**c**). Topography of a SAM for adsorption when the south pole of the magnet is facing the sample (**d**). Magnetic image measured in parallel with the topographic image presented in (**c**).

**Table 1 molecules-25-06036-t001:** Tilt angle analysis of SAMs with respect to the sample normal measured just after adsorption. This table shows the tilt angles of the SAM for both areas with IP easy axis and OoP easy axis and for both cases with an applied external field and without it. It is clear that the tilt angle is correlated with the easy axis direction of the FM layer, even when an external magnetic field is applied.

	In Plane Easy Axis	Out of Plane Easy Axis
No Magnetic field	67.3° ± 1°	62.5° ± 1°
In Plane Magnetic field	69.1° ± 1°	62.7° ± 1°
South Out of Plane magnetic field	-	63.3° ± 1°
North Out of Plane magnetic field	-	60.4° ± 1°

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
