# Peer review of "Correlation between Ferromagnetic Layer Easy Axis and the Tilt Angle of Self Assembled Chiral Molecules"

_molecules, 2020, doi:10.3390/molecules25246036_

Round 1

Reviewer 1 Report

The work entitled “Correlation Between Ferromagnetic Layer Easy Axis and the Tilt Angle of Self Assembled Chiral Molecules” by Paltiel et al. continues the previous series of works developed by Naaman and Paltiel regarding CISS effect and enantiomer separation techniques using magnetic substrates. In this work, with the help of MFM, the authors studied the correlation between the anisotropy of the magnetic substrate and the tilt angle of the adsorbed chiral molecules.

The manuscript is clear without spelling errors. However, there are some issues to be addressed.

Caption of Figures 4 is incomplete.(119 row)

Please change (191 row ) L-Polyalanine into Poly-L-alanine, add thiolated alpha helix and eliminate (192 row) the sequence [H]-CAAAAKAAAAKAAAAKAAAAKAAAAKAAAAKAAAAK-[OH] or specify what each letter represent (C, A, and K represent cysteine, alanine, and lysine).

Please specify in the Materials and Methods what was the magnetic tip used in the MFM measurements.

The Supplementary Materials file is missing.

We recommend the paper for publication after minor revision.

Reviewer 2 Report

By looking into the tilt angle of the self-assembled chiral molecules the authors studied the interaction between the chiral molecules and the ferromagnetic layers, which would definitely benefit the enantiomer separation techniques by using magnetic substances.  This paper was well rewritten and presented in a concise and accurate way.  I agree it could be published in this journal of Molecules. 

A tiny flaw that I found could be a couple of typos: in line 64, Figure 1- should be Figure 1.  

in line 119, Figure 4. and magnetic image could be a magnetic image? 

If the tilt angles could be marked out in Figure 1. it would really help common readers read and understand the paper.

It would also be interesting to run some of the measurements with D-PAL, which might lead to a more convincing conclusion.
